# Managing Wine Tourism and Biodiversity: The Art of Ambidexterity for Sustainability

**Claire Lamoureux** [1,*] **, Nindu Barbier** [2] **and Tatiana Bouzdine-Chameeva** [1,*]

1   Department of Operation and Information System Management, Excellence Center of Food, Wine and Hospitality, KEDGE Business School, 33405 Talence, France
2   Laboratoires Dubernet, 11100 Narbonne, France
*   Correspondence: claire.lamoureux@kedgebs.com (C.L.); tatiana.chameeva@kedgebs.com (T.B.-C.)

**Abstract:** Wine tourism is a mutually beneficial opportunity for customers to experience a wine region and for wine producers to promote their individual practices and approaches in the wine-making process. This article aims to understand producers' perspectives on the challenges they face when trying to protect their wine estate's biodiversity as they develop wine-related touristic activities. The research is based on an exploratory, multiple case study of wine producers, who are protecting their wine estate's biodiversity on in Languedoc-Roussillon region, France. The study reveals the important synergies between biodiversity and wine tourism to increase global sustainability, to bond with customers and to positively impact the wine region. Yet, promoting biodiversity on a wine estate creates tensions on resources and requires investments which are not always highly profitable. Raising awareness about biodiversity is also much needed at both ends of the "producer-customer" relationship. Our results led us to develop an ambidexterity model, adapted to the management of wineries, that concurrently protects their biodiversity and develops wine tourism. We believe these results can be useful for both public and private stakeholders to adapt their wine tourism service offers, and support wine producers in their quest to develop biodiversity of their lands and overall sustainability.

**Keywords:** biodiversity; wine tourism; wine producers; stakeholder; ambidexterity model

## 1. Introduction

Wine tourism is a component of a region's global tourism and, as such, should share the same objectives in terms of communication, diversification of activities, and sustainability. As stated in [1], tourism should preserve the characteristics of a local area, including its environment and ecosystem. Environmental concerns are growing amongst the population and tourism must now support sustainability values [2,3]. Keeping in mind that tourists' preferences shape demand [4], supply needs to evolve to meet sustainability expectations. Tourism is also, by nature a strong leverage tool to develop the economy of a rural area [5]. Therefore, wine tourism plays a key role for local and rural development [6]. Wine regions promoting an eco-friendly approach are more likely to attract visitors with ethical and environmental concerns, [7] and the proportion of such tourists has increased in the past decade [8]. Furthermore, wine tourism has proved to be an excellent way to advertise products and their production methods [9]. Advocating sustainability is also a useful tool for winemakers to interact with customers [10], share knowledge about the production, and build a strong relationship [9]. A vibrant wine tourism experience should be tailored for different tourist interests and profiles ([11,12]) and as such, it is a beneficial business diversification that helps reach new market shares. Wine tourists have diverse profiles [13], thus increasing direct sales thanks to an enlarged audience and potential profitably [14]. Environmentally-aware visitors are better profiled nowadays, and this segment of customers is growing. In 2010, Barber et al. [15] first described them as *"female*

*possessing stronger environmental attitudes, [ . . . ] thus influencing stronger behaviors toward purchase intention*", supporting the idea that the natural beauty and landscape of a region is a main driver of destination choice [16], and that tourists who enjoy these natural and cultural assets tend to spend more on average during their stay [17]. Sigala [18] supports the idea that a multi-dimensional experience, including the socio-cultural elements of the wineries, will conquer wine tourists and inspire in them a deeper feeling of connection with the wine estate, beyond their economic resources. As a result, wine regions have become aware of the benefits resulting from sustainable touristic practices on their environment and on social and economic outcomes for local communities. [19] Consequently, researchers have noticed the emergence of a sustainable wine tourism phenomenon in recent years, involving "the identification and management of unique resources [that] are indispensable for defining a sustainable wine tourism offer" [20]. One could expect that a sustainable wine tourism activity only brings positive outcomes: more tourists, increased sales, better, long-term relationship with customers while respecting wineries' natural environment. However, sustainable wine tourism cannot exist without sustainable farming practices, whereby the wine estate's biodiversity of is preserved and enhanced. Although protecting biodiversity is one the 17 Sustainable Development Goals designed by the United Nations in 2015 (https://sdgs.un.org/, lastly accessed on the 18 October 2022), its implications are not fully understood by neither customers nor producers [9]. Szolnoki [21] reports that many wine producers cannot differentiate sustainable production from organic production. Though these two types of production do not conflict, obtaining an organic label does not suffice in itself to guarantee that biodiversity is preserved and enhanced on the land.

This study focuses on independent wine estates, as they represent the majority of wine production in France (57% of wine volumes are produced by individual wineries (https://www.intervin.fr/etudes-et-economie-de-la-filiere/chiffres-cles, lastly accessed on 1 September 2022)). We shall attempt to contribute to the rather limited studies on wine tourism's and wine estates' sustainable goals [22]. We shall therefore present a model encapsulating the ambidexterity wine producers must develop when they are protecting the biodiversity on their estate and managing their wine tourism business, and the resources they must use to find valuable solutions for their growth strategy.

## 2. Literature Review

The concomitant interest for greener wines from both customers and producers [23] seems like a fertile common ground to develop a strong wine tourism sector that supports sustainable farming practices. A recent Italian study [24] found that customers were willing to pay more for a wine whose certification included a biodiversity component; this was true for any type of wine, entry-level or high-end, of the range. However, the same study showed that the definition of biodiversity differed among the population. In other words, customers adhere to the concept of sustainable winemaking, but they do not know what it entails for farming practices, wine-making processes [25], and impacts on vineyards' biodiversity. Therefore, the need to educate customers and help them understand the scope of sustainability in general, and biodiversity in particular, appears very clearly. The conversation on such a complex topic cannot merely boil down to the issue of eco-labels. At the other end of the spectrum, producers do not actually implement biodiversity protection measures *per se* [9,26], as they solely reduce chemical treatments [9]. A New Zealand study [9] questions the real motives of producers to make more efforts to protect their ecosystem, as it appears that sustainable certification can only be considered as a differentiation strategy in a competitive and customer-focused market.

DeLong [27] gives the following definition of biodiversity: "*Biodiversity is an attribute of an area and specifically refers to the variety within and among living organisms, assemblages of living organisms, biotic communities, and biotic processes, whether naturally occurring or modified by humans.*". Biodiversity is a rich and multi-faceted notion, it is dynamic and evolves around space and time [28]. Not only must it be protected, but it can also be enhanced through the use of appropriate practices. Vine monoculture on a vineyard

represents, in itself, a threat to the wine estate 's living organisms' diversity [29]. We could argue that by having cultivated vines in France for centuries, human's influence has already shaped the biodiversity of the vineyard. Indeed, the definition of *terroir*, dear to the viticulture sector, encapsulates the mediation role of human intervention in the expression of the soil and the climate through the fruit, and the importance it has in the product's identity [30]. Nonetheless, practices such as agro-forestry can improve the diversity of species of the lands more than conventional agriculture and forestry in Europe [31]; weed-control methods such as mulching have also shown good results to protect and increase the soil biodiversity [32]. This proves that farming practice choice can positively impact the vineyard's biodiversity. However, Szolnoki [21] noticed that some wine producers can hardly distinguish sustainable production from organic or biodynamic production. One can thus assume that the same confusion applies to the distinction between sustainability and biodiversity. Therefore, in our study we clearly differentiate biodiversity measures from the wine estate's general sustainability. Regardless of the positive impact of communication on sustainable practices, most producers protect their environment in order to protect their own life quality, [9] since they work and live on their estate. For instance, in France, the findings regarding pesticides' health hazards for all stakeholders impacted the legislation [33] and led to an evolution in treatment use in vineyards. To describe these different motivations, e.g., life quality, respect for the environment, the need for a business differentiation, Casini et al. [34] propose a matrix of four behaviors adopted by wineries towards sustainability (recapped in Figure 1).

- The "devoted": they are great promoters of their winery's sustainability. They implement sustainable practices and communicate on them. They constantly invest in education and training of their staff and customers.
- The "unexploiters": they have adopted sustainable practices, but they do not communicate about them.
- The "opportunists": they massively communicate on the sustainable side of their work in order to differentiate themselves on the market, but they lack a deeper motivation or interest for sustainability.
- The "laggards": they are not interested in sustainable farming at all, and they are not aware of or fail to understand its benefits in terms of communication.

Based on this proposition [34], we wonder whether that is still the case in a more conservative country such as France, which is traditionally more product-driven [22]. As we focus on the importance of biodiversity in this study, we will adapt this classification to the producer's behavior towards biodiversity.

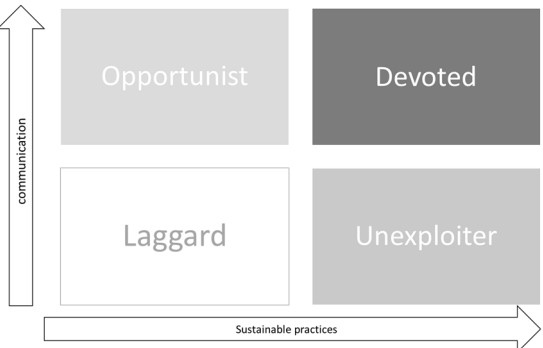

**Figure 1.** Behavior matrix towards sustainability, adapted from Casini et al. [34]. Copyright 2010 by [Casini, Cavicchi, Corsi, Santini]. All rights reserved.

Moreover, the main reason for choosing a wine region destination is the quality of its landscapes; the natural environment [35], and thus its *terroir* components ([36,37]). Hence, one can expect the improvement of biodiversity to have a positive impact on the destination's attractiveness. This is Woodside and Lysonki's assumptions [38]. They have found

that a region's positive image made it more attractive, and in this case that an environmentally friendly reputation would increase its attractiveness for customers, especially regarding eco-certified wineries ([39,40]). However, we do not know if producers are aware of this potential leverage and how they are in fact using it to serve their communication and develop their wine tourism. We would like to contribute to recent studies ([41,42]) to better understand the effects of sustainable wine growing on the quality of the wine tourism experience, from the producers' point of view.

Wine tourism has a direct economic value for the region as it supports local networks of producers beyond wine makers, involving regional food producers [43], hospitality [44], and arts and crafts [45]. Nonetheless, the only large-scale study measuring wine tourism's turnover so far is by Tafel and Szolnoki [46], and it focuses on the German wine regions. At the producer's individual level, exact income is hard to estimate, but wine tourism does increase the number of revenue streams [47] and direct sales ([48,49]). At the heart of wine tourism's economic role lies the complex combination of the wine tourism's capacity to increase brand equity, and this can concern regional brands ([50,51]), consumers' loyalty, and the long-term repetition of visits and purchases.

Lastly, our research assesses whether biodiversity practices and wine tourism activities are compatible on a wine estate. Implementing sustainable farming practices such as organic viticulture can create internal pressures on the winery's management, as producers expect a higher workload for a smaller yield [52]. On the other hand, wine tourism is costly for resources and investment [53]. The balance between these two activities must be found to run a successful, sustainable wine tourism in a sustainable wine estate. If is not already the case, wine estate managers must become increasingly versatile and resourceful to fill more and more diverse roles.

In conclusion, by providing an empirical investigation on the wine producers' own definition of their vinicultural ecosystem and the types of practices they use to protect biodiversity and assess their management tensions, we expect to contribute to the field of sustainable wine tourism by answering the following research questions:

- To what extent do producers understand biodiversity, and how do they implement protection practices on their estates?
- Do these biodiversity protection measures impact their winery's attractiveness?
- Are biodiversity protection and wine tourism compatible in terms of company management?

## 3. Materials and Methods

### 3.1. Research Settings: Languedoc Roussilon, a French Touristic Region

As wine growers are under public scrutiny regarding their societal and environmental duties, they are moving forward and taking measures to attain sustainable development and protect the biodiversity on their vineyard. The French Government is supporting the wine industry's efforts by funding research and industry-wide CSR development programs. For instance, following the 2007 "Grenelle de l'Environnement" [54], the "Haute Valeur Environnementale" label (High Environmental Value) was created in 2012 to certify agricultural units complying with a phytosanitary product strategy, ensuring biodiversity preservation, managing fertilizers, and guaranteeing quantitative water management.

Languedoc-Roussillon is one of the largest producers of organic certified and biodynamic wines in France; four of its *départements (counties)* are in the top ten organic wine-producing *départments* in France, and over 2000 producers work on 38,145 hectares of organic-trained vineyard (source: Agence bio, French national agency for the development and promotion of organic farming [55], lastly accessed on 14 September 2022). As such, Languedoc Roussillon is a pioneer in sustainable wine growing practices in France. Furthermore, it is also the first touristic region in France (https://www.ladepeche.fr/2019/07/03/loccitanie-premiere-region-touristique-de-france,8293106.php, lastly accessed on 21 September 2022), combining a clement climate and a great variety of landscapes with the Mediterranean Sea coastline and the Pyrenees mountains. The region attracted 30 million

tourists in 2019. Its naturally dry, windy, and sunny weather favors the reduction of phytosanitary product use. Nevertheless, as a southern European wine region, Languedoc Roussillon is especially exposed to global warming's effects on wine quality and must question their cultural practices [56]. The same climate change impacts the biodiversity, but as suggested in 2007 [57], the richer the biodiversity is the better it will resist to the rise in temperature. Therefore, this region has invested in sustainable development, and thus appears to be relevant for our case study.

*3.2. Design of the Study and Data Collection*

To address our research questions, the study adopted an exploratory, multiple case study approach, considering fourteen wine companies in the Languedoc region in France (listed in Table 1).

**Table 1.** Typology of the wineries, position of the executive member interviewed, certification and wine tourism offer.

| Interview ID Number | Type of Wine Estate | Vineyard Surface | Position of the Interviewee | Eco-Certification | Wine Tourism Offer |
|---|---|---|---|---|---|
| 1 | Independent winery | 35 ha | Wine tourism manager | European organic certification | Vineyard tour, tastings, workshops, fine dining, concerts, masterclasses |
| 2 | Cooperative of 260 producers | 2000 ha | Commercial manager | Vignerons Engagés | Tastings, visits, workshops, vineyard tours |
| 3 | Independent winery | 70 ha | Winery owner | European organic certification | Tastings |
| 4 | Independent winery | 32 ha | Winery owner | European organic certification, Demeter | Parties, vineyard tours, exhibitions, tastings |
| 5 | Independent winery | 30 ha | Commercial director | European organic certification | Concerts, exhibitions, markets, vineyard tours, picnics |
| 6 | Family-owned winery | 50 ha | Winery manager | ISO 26000, HVE | Open days, vineyard tours, tastings, soft mobility tours, workshops |
| 7 | Independent winery | 60 ha | Winery owner | Terra Vitis | Vineyard tours, child-friendly activities, exhibitions, catering, electric scooters, concerts, escape game, private venue |
| 8 | Independent winery | 50 ha | Winery owner | HVE | Tastings, wine shop |
| 9 | Independent winery | 50 ha | Communication and event manager | European organic certification | Reception venue, exhibition, concerts and DJ sets, guided visits, pub |
| 10 | Independent winery | 23 ha | Winery owner | European organic certification | Corporate workshops, Airbnb Experience, catering, bed and breakfast |
| 11 | Association | 18 ha | Association manager | European organic certification | Vineyard tours, visits, concerts, farmers' market, concerts, exhibitions, wine shops |
| 12 | Independent winery | 150 ha | Wine cellar manager | European organic certification | Tastings, vineyard tours, wine cellar visits, workshops |
| 13 | Independent winery | 60 ha | Winery owner | Terra Vitis | Wine shop, tastings, workshops, picnic baskets |
| 14 | Cooperative of 70 producers | 700 ha | Communication and wine tourism manager | Ecophyto | Electric bike tours, vineyards tours, wine cellar visits, parties, night visits, diners, work seminars |

Nota Bene: When quoted in this article, wine estates are referred as their ID number into brackets.

The two main criteria for selecting wine estates were the following:

- Eco-certification
- Wine touristic activity

Table 1 summarizes the presentation of the wine estate, including their eco-certification and wine tourism offers, as well as interviewees' functions and positions. Each wine estate has been reviewed individually, and the results for each project is recapped below, e.g., the size of their vineyard, interviewees' positions, their eco-certification and a summary of their wine tourism offers.

By looking at these companies' strategies, we analyze the different ways biodiversity is understood and managed in the context of wine tourism. We will also assess the impact of the practices implemented by companies on the wine estate's attractiveness.

This corpus, combined with our database of archival documents, provided extensive information on the company situation, the approaches used in the companies, wine tourism-related events, and other initiatives. We mainly targeted the executive members of wine estates, such as general managers (see Table 1), to collect their ideas on decision-making and their wine tourism business strategy. These participants brought specific expertise relevant to biodiversity issues and company management. We firstly asked them to describe their wine tourism activity, history, motivations, and the place this activity takes in their company's management as well as the time dedicated to it, the human resources it requires, and the profitability it brings. We then asked how they would describe their vinicultural ecosystem and the measures taken to protect it, giving interviewees the opportunity to answer using their own definition of biodiversity. The research could then investigate how different elements of their business strategy contributed to the development of wine tourism. Hence, we also asked interviewees about the link they see between biodiversity and their wine tourism activity, and inquired about the resources they mobilized to manage both. We concluded the semi-guided interview by asking them about their perspectives on the future of their business and the role biodiversity and wine tourism will play in the picture. A detailed interview script is available in Appendix A.

One of the research team members conducted the interviews, either face-to-face, when possible, or via video conferencing between March and May 2020. The first COVID-19 lockdown had just been established in France, and the wine estates managers had not noticed major consequences for the business yet; however, in their interviews, some of them mentioned postponing the development of any new projects. The interviews lasted forty minutes on average. All interviews were recorded and manually transcribed. All interviewees agreed to be recorded and they were guaranteed their interviews would remain anonymous. All interviews were validated with the recordings by the interviewee to ensure data accuracy [58], and they have been granted access to the transcript during any time of the analysis. In total, the research team conducted, recorded, and transcribed fourteen interviews. The interview questions were created in French and translated into English for the manuscript; a similar process occurred for the transcripts. Moreover, authors triangulated data from the interviews with archival sources (for credibility and reliability) to seek converging evidence.

### 3.3. Data Analysis

We performed the data analysis in two stages for data structuring. During the first stage, one research team member assigned their own code to the data gathered to represent a specific meaning, in an inductive approach. During the second stage, the other research team members grouped the codes into categories, as recommended by Miles et al. [59], and the differences were reconciled. Overall, the process followed the principles of theory building identified by Eisenhardt [60] to discover new concepts from the case study. We completed our analysis with a deductive approach, testing each category against the literature [61]. The authors also took several measures to increase the study's reliability and validity: ensuring triangulation of views from the companies with the available data, by using the consistent protocol of data analysis and isolating patterns to identify

commonalities among data collected to establish consistent generalizations across the cases, as suggested by Voss et al. [48]. The reviewed categories were then grouped into larger themes [61] after discussions among the authors.

We used N-vivo software to code transcripts and visualize the result matrix, with lines representing cases, and columns the categories sorted by theme. We then went through a vertical reading to find contradictions between the wineries, and a horizontal reading to highlight contradictions within single interviews. We reported any inconsistent findings or unexpected results in the Discussion chapter. We finalized our interpretation by proposing a model of ambidexterity between Wine Tourism and Biodiversity from the wine producer's point of view.

## 4. Results

*4.1. Data Treatment*

After coding the transcripts, we counted the number of references in each category. We obtained the following matrix of analysis, presented in Table 2.

**Table 2.** Organization of categories and sub-categories into themes of the analysis.

| Categories and Subcategories | Theme | Number of References * |
|---|---|---|
| Activities | | |
| Managerial consequences<br>- Revenues/Profitability<br>- Investments/costs<br>- Human resources | Wine Tourism | 261 |
| Motivations | | |
| Certifications<br>- Tensions | | |
| Biodiversity:<br>- Practices<br>- Definition<br>- Investments<br>- Incomes | Sustainability | 208 |
| Values | | |
| Communication<br>- Biodiversity communication<br>- Educating visitors<br>- Wine Tourism communication | Customer relationship | 156 |
| Attractiveness | | |
| Visitor profile | | |
| Synergies<br>Tensions | Relations WT/Biodiversity | 75 |

* The number of references was aggregated from each category.

*4.2. Biodiversity Knowledge Is Contrasted among Winery Executives*

To be addressed, this first research question on the definition of biodiversity must be decomposed into interdependent elements: the importance of biodiversity compared to overall sustainability, the motivation to implement biodiversity friendly practices, and eco-certification issues.

First, all of our interviewees showed a genuine interest in protecting the environment, and they were all able to mention practical measures implemented to enhance the overall sustainability of their estate. Supporting [9], all interviewees reported their conviction to do good for the environment by protecting their biodiversity, respecting their values, their health, and their quality of life. However, only nine of fourteen (64%) of them were able to give a specific definition of biodiversity. Most of them considered the protection of biodiversity to be included in their certification, but as we have seen before, the different eco-labels have various levels of expectation in terms of biodiversity protection and/or development. The wine estates based in a Natura 2000 Zone and HVE certified were the most aware of what biodiversity is and how it can be measured; their definition of biodiversity is accurate, and they know how to observe it on their site. They must audit the number of species on a regularly basis due to these certifications. Some will argue that any practice taken to contribute to the global sustainability of a winery will protect their biodiversity, nonetheless, it will not necessarily increase its development. Hence, "collecting rubbish and recycling", which is often quoted as an effort to protect the environment, will reduce the damage of the human presence on the natural site, but will not increase the diversity of species.

Adapting Casini et al.'s [34] classification of biodiversity, all fourteen wine estates are convinced of the importance of protecting their biodiversity, and twelve of them (85%) were able to name at least one practice directly impacting biodiversity. Out of these twelve, a quarter mentioned their difficulties communicating on these measures, either because they use their certification instead, or because they do not want a fundamental value to become a commercial argument. One interviewee demonstrated some contradictions, describing the importance of the natural environment but wanting to develop a wine tourism project involving classic vehicles and 4 × 4 vehicles. Therefore, we could classify these wine estates as follows:

- Laggard: none
- Opportunist: 1 winery (1)
- Devoted: 9 wineries (2, 4, 5, 6, 8, 9, 10, 11, 12)
- Unexploiter: 3 wineries (3, 7, 13)

Winery owners' opinions on the different certifications are contrasted: some interviewees even reported them as conflicting and damaging to the quality of the communication. If all of them are proudly certified, their motivations can be divided in two categories: those using the certification as a way to be audited and improve their biodiversity, and those using the certification as a mean of recognition of their work and as a communication tool; 71% of the interviewees reported difficulties in obtaining these certifications as "*getting certified is also a lot of administration, time and money.*" (8). They can even get discouraged "*a few years ago, we started the process for HVE certification, but it was very complex from an administrative point of view, so we did not finish the process.*" (9). 35% of winery executives described consumers confusion in the face of the many certifications, their meaning, their absence of communication: "*it is still vague for them, because wine and organic products regulation is very complex.*" (12). "*Educating visitors on the different labels and the different approaches is complicated.*" (8). We illustrated this confusion in Table 3 by ranking the different levels of expectations related to biodiversity for each label used by the interviewed wineries. This table is an adaptation of official sources on biodiversity indicators, obtained from certificating organizations' websites.

**Table 3.** Impact on biodiversity of each certification and official sources of the certification organism.

| Eco-Labels | Official Source | Impact on Biodiversity | Methodology | Comments |
|---|---|---|---|---|
| Terra Vitis | Certification website https://www.terravitis.com/ | 80 points of audit controlled every year (no specific criteria detailed). For biodiversity: development of living soils, maintenance of fences, forest, vegetal covers, limiting the treatments | Quantitative and qualitative indicators | The most demanding biodiversity certification with yearly audits of the species |
| HVE Haute Valeur Environnementale | French national agriculture website https://agriculture.gouv.fr/ | Sesquiannual biodiversity audit: insects, trees, hedges, grass strips, flowers . . . (no specific criteria detailed) | Quantitative and qualitative indicators | The second most demanding biodiversity certification with audits of the species every 18 month. |
| Vignerons Engagés | "vignerons engages" Certification website https://vignerons-engages.com/4-piliers-et-12-engagements/ | "Biodiversity diagnoses of the territories [ . . . ] actions to protect endangered species, such as setting up beehives, insect hotels, or nesting boxes. [ . . . ] multiply positive cultural practices for the life of the soil and for the creation and maintenance of animal and plant species [ . . . ]." | Quantitative and qualitative indicators | Biodiversity is addressed by the variety of landscapes and the importance of sheltering the different species of livings. Frequency of diagnoses is not specified. |
| Demeter | Demeter website: https://www.demeter.fr/biodiversite-2/ | "10% of the useful agricultural surface of the farm is dedicated to areas of biodiversity" | Quantitative indicators | Biodiversity preservation does not concern the whole wine estate, it only applies to a percentage of the surface. |
| ISO 26000 | AFNOR (French Agency of norms) website https://bivi.afnor.org/ | Norm NF X32-001 Biodiversity audit, creation of indicators, measure and communication (no specific criteria detailed) | Quantitative indicators | The indicators are not specified on the official source and the process remains opaque. |
| European Organic label | French national agriculture website https://agriculture.gouv.fr/ | "Excludes the use of synthetic chemicals, GMOs and limits inputs." | No indicators of biodiversity | Biodiversity per se is not addressed, only implied in the reduction of authorized chemicals and the interdiction to use GMOs. |
| Ecophyto | French national agriculture website https://www.mesdemarches.agriculture.gouv.fr/ | Certification of knowledge and use of phytosanitary products, no mention of biodiversity | No indicators of biodiversity | Biodiversity per se is not addressed, concerns only the optimization of phytosanitary treatment use. |

All websites have been lastly accessed on 21 September 2022.

Certifications can be costly and time-consuming for producers, even when they have been implementing biodiversity practices for a long time. Without a powerful communication from the labels on their actual effects on biodiversity, producers themselves can feel discouraged and can struggle to educate consumers, thus substituting themselves to the diverse certification organizations. As a result, consumers cannot make an informed choice. A widely recognized label can be motivation to get certified: "*In 2014, we had the AB logo. A little late because my father did not believe in the label. The future proved him wrong because today the label is showing its strength.*" (10). It is important to mention that no interviewee reported to have deeply changed their farming practices to get certified. This means they chose the label that best matched their values and their own interpretation of biodiversity protection. As it was previously mentioned, there was no sign of opportunism in the use of the certification.

*4.3. The Unexpected Educationnal Role of Wine Tourism*

Wine tourism allows wine estates to improve their reputation or seize the opportunity offered by a region's touristic flow. These improvements of direct sales and recognition are the first reasons to start a wine tourism activity, as shown in Table 4.

**Table 4.** Motivations to start a wine touristic activity.

| Motivation | "Quotes" (Wine Estate ID Number cf. Table 1) |
|---|---|
| Increasing sales | "To boost the direct sale of wines, but also so that visitors come to the peninsula to discover our wines and eat at the restaurant." (11) |
| Increasing number of visitors and customers | "We thought that wine tourism would bring people here and make us known, especially to locals". "We want to develop wine tourism because we think it is important to be able to offer the most varied offer possible to attract new customers." (9) "Our estate is suited for wine tourism and it would be a shame not to use it." (5) "The goal was to develop our customer base" (14) "The [location] attracts nearly 200,000 visitors a year, [ . . . ]. So we opened a wine shop and a restaurant which exclusively serves wines from the estate." (11) "Bring people to our location" (7) |
| Reputation | "[We want to] organize a lot of events to increase our "popularity"" (1) "Our goal was to strengthen our reputation" (4) |
| CSR goals | "Wine tourism makes sense, it is not only about diversifying our offer, it meets our CSR ambition", and "More than anything, wine tourism enables us to meet qualitative objectives in our CSR goals." (6) |
| Presentation of the wines | "It allows us to present our wines in a festive way" and "It allows us to showcase our wines and the products of our restaurant" (1) "We also organized a Christmas market to present our wines." (11) |
| Increasing their service range | "We want to develop wine tourism because we think it is important to be able to offer the most varied offer possible to attract new customers." and "We want to develop wine tourism with the construction of accommodation on site: this would allow us to have a global offer." (9) |

Wine estates' main motivation to start the wine tourism business is to increase their visibility and attract new customers and subsequently increase sales and reputation. Yet, with the evolution of wine tourism through time, one can also see a growing divide between wine estates who have decided to invest in wine tourism to diversify their business and generate an additional income, and those who are using it as a communication tool only; to create an opportunity to meet their customers and discuss their practices with them. For the former, wine tourism is becoming profitable, though not it is not their main source of income: "*Last year, we made €15,000 exclusively from wine tourism, so that's pretty good. And then it pays off in terms of additional sales and reputation . . . *" (10). For the latter, the profitability of the wine tourism business is not the main goal, they try to limit losing money on their investment by charging a small fee for the activity: "*the 5€ fee is used just to cover the investment*" (5), but the main income is generated by wine sales.

Despite these different conceptions of wine tourism, which can be explained by internal factors such as financial availability and external factors such as location and wine estate layout, they all agree on the fact that their touristic activity helps them reach a new customer base. Thus, half of them have noted a rejuvenation of their visitors, "*In recent years, [our visitors' population] has been getting younger. We have more women too. And young couples. I have the impression that there are also more and more French people: people are reclaiming their terroirs*" (5). This new target is attracted by the natural environment of the wine estates; 57% of interviewees mention the importance of preserving their natural environment as part of the attractiveness of their winery: "*A natural environment can be an element of differentiation for a winery if we take care of it.*" (6), and a third of these mention the organic label as a tourist's motivation to visit them in order to know more about their practices. "*We have a lot of visitors who come to us because we e farm organically. This is a big element of differentiation compared to a conventional vineyard*", (5). In conclusion, biodiversity protection can be a convincing argument to generate more visits to a wine estate and reach a new panel of consumers with an interest in the environment and ecology.

Though wine producers have been implementing biodiversity preservation practices in the vineyard for years, showcasing them was not their first intention when starting wine tourism. The opportunity to exchange views with visitors and raise their awareness on the biodiversity of the wine estate is a positive outcome of wine tourism. The model below (Figure 2). is adapted from our analysis, and shows the articulation of the visitor relationship.

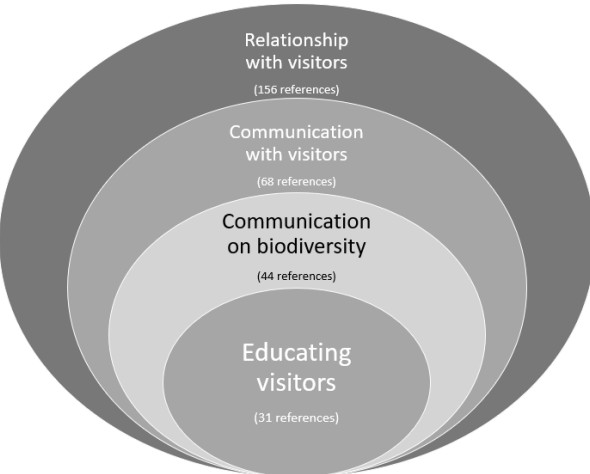

**Figure 2.** Model of "Relationship with visitors" theme.

Educating visitors is mentioned in 85% of the interviews and it represents 70% of the references made about communication on biodiversity. This supports the idea that biodiversity requires explanations and information to be fully understood, and shows how wine producers use wine tourism to inform the visitor, and that the first step to communicate about biodiversity is to educate the public. Figure 3 is a word cloud of the twenty most frequently used words in the category "Educating visitors", showing the diversity of shape and form that communication can take: describing their biodiversity with signage in different areas of the vineyard explaining their farming practices wine-making processes, or raising general awareness about sustainability of the wine estate as a whole set of interactions.

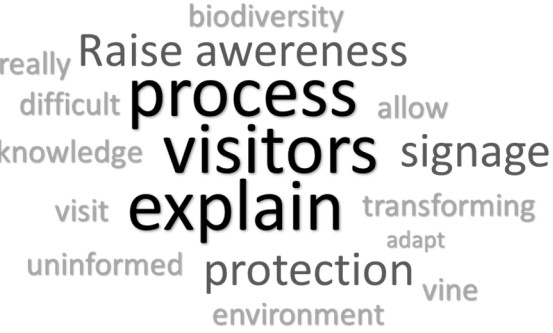

**Figure 3.** Word cloud for the "Educating the visitors" sub-category.

This educational approach seems to "*reassure, question [the customer]*" (1). Interviewed wineries' farming practices were sustainable long before wineries communicated on the topic and obtained their certification (as said in Section 4.2). Sales and reputation were the main drivers for starting a wine tourism business, but producers quickly realized the positive consequences of engaging with visitors to educate them. It became a unique way to promote their work and biodiversity, as it provided customers with information and understanding about their certifications and improved public knowledge on these issues.

However, we reported conflicting views among our interviewees. For some "we can entertain people and educate them at the same time, it goes together." (5). Others found it important to remain "accessible [ ... ] If we get carried away in great technical explanations when they are there to have a good time with friends, they will not like it." (14).

*4.4. Managing Wine Tourism and Biodiversity: When Tensions and Synergies Overlap*

Whether it stems from personal beliefs or it is used for differentiation purposes, protecting biodiversity is part of the wine estate's management strategy: "*Protecting biodiversity is obviously a major focus of our investments, of our reflection*" (5). Five interviewees admitted that protecting biodiversity was an investment: "*[protecting e biodiversity] has a cost and I understand that this can restrain producers with limited financial means*" (4). Nevertheless, profitability is not their goal: "*For example: we calculated that not using herbicides cost us 15,000 euros (labor in the vineyard etc. ... ). Maintaining agro-ecological infrastructure costs me 17,000 euros per year, etc. [ ... ] Maintaining and protecting the landscape is costly, but it's worth it!*" (13). Their conviction justifies the expenses and they have the feeling it supports the commercial development of the winery: "*I am convinced that it makes people buy more and more, and that it reassures the consumer.*" (1). Wine estate 3 considers that "*it impacts external sales*" and "*brings slight changes to the type of customers*".

Synergies between wine tourism and biodiversity are not limited to the attractiveness of the site. For wine producers who are concerned about their environment and eager to share their passion, they really are two sides of the same coin. "*The protection of biodiversity gives a meaning to wine tourism*" (6). Wine tourism becomes a public demonstration of the environmental values of the wineries for 85% of the interviewees, as long as the activities are consistent. "*We want the activities to be really in line with our identity and our values. We are looking to develop "green" visits: i.e., by electric mountain bike.*" (12). Thus, they show great care in guaranteeing their tourism offer's sustainability through the smallest details: "*we only have hard cutlery for catering, we use reusable plastic cups for drinks.*" (7). Wine tourism is also a powerful way to engage with visitors and consumers. "*Wine tourism allows us to explain our practices to people: how much detail we go into regarding procedures depends on their level of interest.*" (8). The limits to this approach are that wine tourism must remain a pleasant experience; a form of entertainment, not a lesson. Additionally, this approach cannot target mass tourism "*It is important to adopt an educational approach but also to guarantee entertainment: visitors should not feel too constrained. [ ... ] Tourism can be a threat to biodiversity, so knowing how to manage flows, educating visitors is necessary... The balance must be found*" (11). Table 5 recaps the main synergies and tensions between biodiversity and wine tourism found in our analysis.

Besides this effort, wine tourism impacts company management. Human resources are the main issue. All interviewed wineries reported that the extra workload generated by wine tourism required hiring dedicated employees, seasonal workers, or interns. That includes family wine estates wishing to develop their wine tourism business and who must consequently consider opening a full-time position to support this. Alternatively, they must redesign the activity so it can run autonomously: "*We are not a multinational company, resources are limited, especially human resources. So, we were looking for an activity that allowed us to highlight our work without having to recruit someone*" (13).

Wine tourism investments mainly involve renovating or building real estate on the winery site: "*we built low walls, replanted the lawns, bought tables and chairs*" (4) as well as signposting in the vineyard. As wine tourism projects get bigger, investments increase and can lead to the creation of venues for seminar or receptions: "*We have a project to renovate a room [ ... ] to add to the meeting room and the dining hall*" (7). Some managers get even more innovative, with virtual tours and escape games. Wine estate 5 invested on "*the interactive tour, to show that you are active.*" On the other hand, wine estate 10 wants to keep on investing "*because we must always be one step ahead. We thought of "wine gaming" inspired by escape games.*" The more activities they offer, the wider audience they can attract: "*We have a lot of families or young couples who visit our estate, especially the botanical trail. For*

*the tasting activities, customers are older or young couples.*" (6). Another winery even made the following statement: "*For example, if someone comes, with a group, and doesn't like wine, that person should be able to think 'I don't like wine but I had a good time anyway'.*" (7). This demonstrates some wine estates' professionalization with regards to wine tourism as well as the way they use it to widen their audience outside wine consumers, and it also shows how much investment they are ready to make.

**Table 5.** Summary of the quotes mentioning synergies and tensions between biodiversity and wine tourism.

| Theme | Synergies | Tensions |
|---|---|---|
| Site attractiveness | "The attractiveness of a wine tourism site is strongly connected to the protection of biodiversity" (1)<br>"The natural environment is really is an important element of a wine tourism offer" (2) | "[Our] wine cellar is located in a magnificent natural environment but it is rather isolated, so the access is quite difficult." (13)<br>"Mass tourism is a danger" (1)<br>"The number of people must be limited, especially during vineyard visits." (1) |
| New customers | "Preserving biodiversity can attract tourists who want to take part in environmental protection." (11)<br>"We are developing green activities to attract visitors who are sensitive to our cause, who will then become ambassadors for our domain." (10)<br>"People are more and more interested in [biodiversity], we can clearly see that." (12)<br>"In recent years visitors' interest in environmental protection has increased I think visitors will become actors who are increasingly respectful of the environment." (9) | "When we hold events, we invest in trash cans, but people are very disappointing. So, the next day, we pick up trash" (2)<br>"It's great to put trash cans and signage everywhere to educate visitors about protecting biodiversity, but honestly it depends on everyone's good will! "(12). |
| Wine tourism activities | "For wine tourism, we would like to set up signs that provide information about wildlife, depending on the biodiversity diagnosis" (2)<br>", There are signs on the botanical trail to explain the interaction between the environment and the vines" (6).<br>"We consider all our activities to be eco-friendly, including visits and tastings." 10 | "We promote soft mobility: exploring the vineyard in a 4 × 4 would make no sense, handing out goodies from China at the end would be totally stupid too." (11)<br>"As soon as we create a new activity, we must try to have the minimum impact possible on the landscape and our environment: use clean materials, do not damage nature." (9). |
| Communication | "Green wine tourism enables us to improve the image of agriculture: we show that our vineyard is also our home, and that there is no greenwashing." (14)<br>"It is not an activist wine tourism, it is rather a delicate, chilled, educational wine tourism. "(13)<br>"Wine tourism allows us to explain our practices to people: how much detail we go into regarding procedures depends on their level of interest." (8) | "People do not come to us because we do green wine tourism activities. We are not known for that, even if we are pioneers" (14).<br>"It's really focused on showcasing the wine, rather than showcasing the natural environment." (2)<br>"We are not here to teach people. It is difficult to constrain visitors. "(2)<br>"[ . . . ] Manage to raise awareness while not transforming these protection procedures into constraints." (7) |
| Values | "Protecting biodiversity gives meaning to wine tourism." (6) | "Everyone wants to be called "green" but no one makes the effort and they prefer to stay in their comfort zone. Talking the talk is easy, but walking the walk is harder" (2)<br>"Our activities are really based on entertainment, there is no special awareness-raising and accountability with regards to the environment. "(2)<br>"You need to have an inner conviction, and to be able to financially support your convictions."(13) |

*4.5. Model of the Producers Ambidexterity between Biodiversity*

Preserving biodiversity supports the development of a flourishing wine tourism. Conversely, wine tourism—when the offer is sustainable—can provide a strong push for

biodiversity measures. However, due to limited human and financial resources, these two poles can frequently limit each other or compete one with another.

Summarizing our previous results, Table 6 is an adaptation from O'Reilly and Tushman [62]'s ambidextrous leadership matrix, and introduces the model of ambidexterity for sustainable development needed by wine producers to manage their biodiversity and wine tourism businesses.

**Table 6.** Biodiversity and wine tourism ambidexterity matrix.

| Alignment of | Biodiversity | | Wine Tourism | |
|---|---|---|---|---|
| | **Exploitation** | **Exploration** | **Exploitation** | **Exploration** |
| Strategy | Protection of the natural environment | New practices i.e., agroforestry, implementing agro-ecology | Business diversification Development of the winery | Innovation supported by offering new activities |
| Asset | Increasing landscapes' value | Making the vineyard more accessible | Natural landscape as an element of differentiation | Attracting eco-friendly tourists |
| Certification | Administrative cost Financial costs | Biodiversity audit | Sustainable activities: soft mobility, low carbon footprint | Towards a sustainable tourism recognition |
| Communication | Information of the public | Finding a balance between education and entertainment | Cross-industry network External communication | Customer loyalty |
| Company management | Optimizing human and financial resources | Staff training Fund raising | Dedicating human resources Investing in structures | Supporting innovation |
| Rewards | Encouraging wildlife's development | Implementing biodiversity practices Increasing number of species Resisting better to global warming. | Increasing the number of visitors | New income streams and focus on increasing income |

This model is the result of our exploratory study, and should be further tested and finalized in quantitative studies.

*4.6. Discussion*

Lastly, we would like to open the discussion on a point raised by interviewees and suggest a new research topic: the interdependence between a winery and its region and stakeholders. The development of a sustainable wine tourism and the protection of the environment are both beneficial to the region and its actors (local producers, local hospitality industry, local tour operators, etc.) Tensions are detrimental to all of them, resulting in a loss of revenue, reputation, and attractiveness. Instead, all stakeholders would benefit from the synergies. As such, financial aspects must be supported by local, regional, and national organizations. Governmental actors could be involved in the administrative process of certification to enhance eco-friendly wine estates' visibility, while the certification organizations should endorse more responsibilities in communications about their label.

Stakeholders' positions on sustainable practices—especially biodiversity and wineries' CSR impact—necessitate further studies on wine sector investments within the framework of the recent concept of "impact investing" [63]. The involvement of all stakeholders, producers, employees, and customers is necessary to raise general awareness [21].

**5. Conclusions and Recommendations**

The study presented in this paper makes several contributions to recent literature on the interplay between wine tourism and biodiversity. We provide one of the first empirical

examinations focusing directly on this relationship's ambidexterity. Our results demonstrate how wineries define biodiversity as well as the various practices they implement to support their sustainable development strategies. We also show the necessity of a global communication policy about wine industry actors to educate producers and consumers on biodiversity and related measures. Most importantly, we highlight the educational role of wine tourism to start this conversation, placing wine tourism in the framework of edutainment [64]. Finally, we investigated existing tensions that need solving, and synergies that must be developed in order to manage the resources wine producers are using to enhance biodiversity in their estates, thereby increasing their local area's general attractiveness of. As a result, we introduced an ambidexterity model to measure the relative weight of these factors in the wine company's management. Future research may consider how small wine estates could achieve greater success in pursuing their sustainability objectives, combining biodiversity and wine tourism more efficiently, and basing their activities on edutaining practices. Future research should also examine how raising wine consumers and tourists' awareness of biodiversity could contribute to reaching these goals.

The scope of our study is limited, since the observation was only made in one touristic region of a single country. However, this choice reflects the classic situation for this field. Furthermore, we present an in-depth and qualitative investigation, whose findings are rich and likely significant for further theory development in this regard.

Lastly, we would like to start a conversation on the role of cross-industry networks, of local areas' actors and financial stakeholders to reflect upon the possibility to share the investment risks taken by wine producers when they develop sustainable wine tourism that is beneficial to all. Future research should examine the role of regional leadership in helping to achieve better outcomes in sustainable wine tourism.

**Author Contributions:** Conceptualization, T.B.-C.; methodology, T.B.-C.; validation, T.B.-C.; formal analysis, C.L. and N.B.; data curation, N.B.; writing—original draft preparation, C.L. and N.B.; writing—review and editing, T.B.-C.; supervision, T.B.-C.; project administration, T.B.-C. All authors have read and agreed to the published version of the manuscript.

**Funding:** This research was partly funded by French VitiREV project. VitiREV is a project supported by the French government and aimed on the agroecological transition to a new viticulture model, responsible for the environment. There is no financial number (as ANR or ERC) as it is a regional project.

**Institutional Review Board Statement:** The study was conducted in accordance with the Declaration of Helsinki, and approved by the Ethics Committee of KEDGE Business School, on the 16 November 2022. The methodology and the subject studied respect Kedge's values and ethics; the data does not require RGPD processing.

**Informed Consent Statement:** Informed consent was obtained from all subjects involved in the study.

**Data Availability Statement:** Anonymous verbatim of the interviews are available on request.

**Acknowledgments:** This research was partly conducted under the support of the French VitiREV project. Any opinions, conclusions or recommendations expressed are those of the authors and do not necessarily reflect the view of the French Ministry of Agriculture.

**Conflicts of Interest:** The authors declare no conflict of interest.

## Appendix A. Script of the Interview

This interview will last approximately 30 min to an hour and will remain confidential. With your agreement, we are recording it to transcribe it in its entirety.
The wine tourism activity of the estate:

- Can you tell me about your wine estate and your wine tourism offer?
- How long have you been involved in wine tourism?
- Why did you develop wine tourism in your area?

- What are your different wine tourism activities? (Visits, catering, accommodation, events … )
- How much time do you devote to each wine tourism activity, per day or per week?
- How many members of your team/family take part in your wine tourism activity(ies)?
- Did you have to invest in your activity?
    - buildings (construction, renovation, upgrading)
    - landscape (signage, maintenance, wine route)
    - hiring of a dedicated person?
- Do you think this activity is profitable?
- Do you think it attracts a population that would not have bought your wines otherwise?

    How important is biodiversity and its preservation to you?
    What general steps do you take to protect your wine estate's biodiversity?
    Relationship between biodiversity and wine tourism:

- In your opinion, what is the link between biodiversity of a site and its wine tourism activity?
- What role does the natural environment play in a wine tourism offer?
- Is it an asset and/or a constraint?
- What differentiates your wine tourism offer from a "conventional" estate? Values?
- Are your biodiversity conservation initiatives part of your marketing and communication strategy?

    Development perspective:

- In your opinion, how can the preservation of biodiversity be improved when developing a wine tourism activity?
- How can these preservation actions be highlighted for visitors?

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
