# Peer review of "Managing Wine Tourism and Biodiversity: The Art of Ambidexterity for Sustainability"

_sustainability, doi:10.3390/su142215447_

Round 1

Reviewer 1 Report

Dear Authors,

This is a very interesting and emerging topic. Generally, it is well-written but I have pointed put some errors which need to be rectified. Please put this paper through both Microsoft word and Google docs to find any further mistakes once complete. I have also requested you redo your coding for consistency. 

Page 2: Line 14: not sure if solely females with this attitude anymore.

Page 2: Line 52: 'seen emerging' should be 'been emerging' or re-write.

Page 2: Line 75: 'a strong wine tourism' add 'sector or industry' after tourism.

Line 2: 87: The same New Zealand should be "A New Zealand study with similar questions....'

Page 4: Line 175: 'with more 4 departments in the top 10 of the organic wine producing department in France'. I do not know what this means, please re-phrase.

Methods: Need to add a sentence or two of length of interviews without looking at Appendix A as 30 minutes seems too quick for the number of questions. Please acknowledge that the interviews took place at the start of a global pandemic (Covid-19) and if wineries were affected as the world closed at this data collection period. Were the wineries affected by Covid-19? How were the results validated? You gave a transcript to participants to re-read and approve?  Who's ethics application did you follow?

Is this part of a larger PhD study or one part of the French VitiREV project where more studies have been published or to published in the future?

Pages 8-14: Need to be consistent with coding.  eg choose either (11) or (wine estate 11). This will need to done in both written sections and tables.

Page 10: Line 35: 'be consistent with spacing "A.... or "A... and full stops (Line 351 and 352 for example).

Page 10: Line 362: (figure 2.) should be (Figure 2.)

Page 11: Line 370: Figure3 should be Figure 3

Table 7 pages 13-14 - Have the font size increased in bold on the left?

Page 14: Line 466: Re-write 'However, any tensions which many arise can be ......damaging to all of them...'

I would also consider adding a line or two about impacts of climate change on biodiversity in future studies and any new wineries should consider biodiversity at the embryonic stage of winery development as many French wineries have been cultivated for centuries and I image the biodiversity has somewhat changed over time.

Good luck.

Reviewer 2 Report

This is a “trendy” topic and the paper is a nice addition to the expanding knowledge related to biodiversity, sustainability and wine tourism. I only have a few minor comments and suggestions:

I am missing one thing in the first part of the article. The authors write extensively about biodiversity but the do not give much examples of how should the biodiversity of a wine estate be preserved and developed? Can you give a good example? What is considered as good practice?

In the introduction section you write about what attracts wine tourists. You mention various elements but not so much the concept of terroir. Please take a look at the following paper which mention the role of terroir as a motivating factor in modern-day wine tourism and include this element in the introduction as well:

Tomić, N.; Koković, J.; Jakšić, D.; Ninkov, J.; Vasin, J.; Malićanin, M.; Marković, S.B. Terroir of the Tri Morave Wine Region (Serbia) as a Basis for Producing Wines with Geographical Indication. Geogr. Pannonica 2017, 21, 166–178.

Goran Jević, Jovan Popesku, Jelena Jević, Analysis of motivating factors for visiting wineries in the Vršac wine region (Vojvodina, Serbia), Geographica Pannonica, 10.5937/gp24-22781, 24, 1, (56-66), (2020).

Vukojević, Dajana, Nemanja Tomić, Nenad Marković, Branislav Mašić, Tijana Banjanin, Radomir Bodiroga, Tijana Đorđević, and Miloš Marjanović. 2022. "Exploring Wineries and Wine Tourism Potential in the Republic of Srpska, an Emerging Wine Region of Bosnia and Herzegovina" Sustainability 14, no. 5: 2485. https://doi.org/10.3390/su14052485

Congratulations on a nice article!
